# Circulating miR-499a-5p Is a Potential Biomarker of *MYH7*—Associated Hypertrophic Cardiomyopathy

**DOI:** 10.3390/ijms23073791

**Published:** 2022-03-30

**Authors:** Natalia Baulina, Maria Pisklova, Ivan Kiselev, Olga Chumakova, Dmitry Zateyshchikov, Olga Favorova

**Affiliations:** 1National Medical Research Center for Cardiology, Laboratory of Functional Genomics of Cardiovascular Diseases, 121552 Moscow, Russia; pisklova.m.v@gmail.com (M.P.); kiselev.ivan.1991@gmail.com (I.K.); chumakovaolga@bk.ru (O.C.); dz@bk.ru (D.Z.); olga_favorova@mail.ru (O.F.); 2Laboratory of Medical Genomics, Pirogov Russian National Research Medical University, 117997 Moscow, Russia; 3Moscow Healthcare Department, City Clinical Hospital 17, 119620 Moscow, Russia

**Keywords:** hypertrophic cardiomyopathy, pathogenic genetic variants, RNA-seq, miRNA, miR-499-5p, biomarker, *MYH7*, *MYH7B*

## Abstract

Hypertrophic cardiomyopathy (HCM) is the most common inherited myocardial disease with significant genetic and phenotypic heterogeneity. To search for novel biomarkers, which could increase the accuracy of HCM diagnosis and improve understanding of its phenotype formation, we analyzed the levels of circulating miRNAs—stable non-coding RNAs involved in post-transcriptional gene regulation. Performed high throughput sequencing of miRNAs in plasma of HCM patients and controls pinpointed miR-499a-5p as one of 35 miRNAs dysregulated in HCM. Further investigation on enlarged groups of individuals showed that its level was higher in carriers of pathogenic/likely pathogenic (P/LP) variants in *MYH7* gene compared to controls (fold change, FC = 8.9; *p* < 0.0001). Just as important, carriers of variants in *MYH7* gene were defined with higher miRNA levels than carriers of variants in the *MYBPC3* gene (FC = 14.1; *p* = 0.0003) and other patients (FC = 4.1; *p* = 0.0008). The receiver operating characteristic analysis analysis showed the ability of miR-499a-5p to identify *MYH7* variant carriers with the HCM phenotype with area under the curve value of 0.95 (95% confidence interval: 0.88–1.03, *p* = 0.0004); sensitivity and specificity were 0.86 and 0.91 (cut-off = 0.0014). Therefore, miR-499a-5p could serve as a circulating biomarker of HCM, caused by P/LP variants in *MYH7* gene.

## 1. Introduction

Hypertrophic cardiomyopathy (HCM) is the most common inherited myocardial disease, disseminated worldwide with a prevalence of 1:200–1:500 [1], and associated with increased risk of arrhythmogenic sudden cardiac death and progressive heart failure, including in the young. It is characterized by the development of left ventricle (LV) hypertrophy in the absence of obvious causes for the observed magnitude of LV wall thickness [2]. The main concern in studying HCM is its extremely high genetic, morphological and clinical heterogeneity. Approximately 60% of HCM-phenotype presenting patients have pathogenic/likely pathogenic (P/LP) variants in sarcomere or sarcomere-associated genes [3]. About 80% of the P/LP variants in these genes are detected in *MYH7* and *MYBPC3*, which encode β-myosin heavy chain and myosin-binding protein C, respectively [4,5]. Other HCM causing variants are localized in the genes of thin myofilament (*ACTC1*, *TNNT2*, *TNNI3*, *TNNC1* and *TPM1*), Z-disk (*ACTN2*, *CSRP3*, and *MYOZ2*) and thick myofilament (*MYL2*, *MYL3*, *MYH6*) [6]. Most current investigations are focused on discovering the mechanisms of the high clinical heterogeneity of the disease even among the same mutation carriers and the members of one family including monozygotic twins [7,8,9,10,11]. Altogether, these findings are consistent with the emerging opinion that HCM should not be considered solely as a monogenic disease; its final phenotype may be determined by complex intergenic interactions and by the involvement of gene expression regulation mechanisms at various levels [12].

MicroRNAs are small non-coding RNAs that form a coordinated regulatory system and control a variety of genes involved in fundamental biological processes, such as tissue differentiation, proliferation, apoptosis, stress response, etc. [13]. The sequence-specific regulation of targeted mRNAs by miRNAs is one of the most important mechanisms of the realization of hereditary information, which can make a significant contribution to the penetrance of genetic variants associated with HCM and thus in the general heterogeneity of disease phenotype. In perspective, miRNAs could be used as promising biomarkers to predict and monitor the pathologic processes in HCM. In recent years, special attention has been given to circulating miRNAs in plasma as accessible biological material. Circulating miRNAs have been found to be associated with cardiovascular diseases such as heart failure and LV hypertrophy, atrial fibrillation and also genetic cardiomyopathies [14]. To date, several studies investigated circulating miRNAs in HCM using the candidate approach [15,16,17,18,19,20] and identified 19 miRNAs that were dysregulated in disease. Of them, miR-145-5p, -133a-3p, -143-3p, -26a-5p, -29a-3p, 199a-5p, -27a-3p, -199a-3p, and 155-5p were considered to be used as potential HCM biomarkers (reviewed in [21]); however, individually they were characterized by moderate diagnostic accuracy and were mostly nonspecific [21]. Fourteen serum miRNAs were associated with fibrosis; however, the correlations were mostly poor (r < 0.4) [16,17]. Levels of circulating miR-27a-3p, miR-199a-5p, and miR-29a-3p were associated with maximum wall thickness [16], and miR-29a-3p and miR-146a-5p were increased in the plasma of patients with left ventricular outflow tract (LVOT) obstruction [19,20]. These data bring up an issue on conducting large-scale studies investigating the miRNome profile to identify novel biomarkers, which could increase the accuracy of HCM diagnosis and possibly improve understanding of its phenotype formation. In order to make gains in this field we performed RNA-seq profiling of circulating miRNAs in the plasma of HCM patients and individuals without cardiovascular diseases and investigated the diagnostic accuracy of identified miRNAs.

## 2. Results

The comparison of circulating miRNA profiles in the plasma of HCM patients (five females, four males; mean age 50.2 ± 10.3 years) and controls (three females, three males; mean age 31.3 ± 5.9 years) revealed 35 miRNAs, the level of which significantly differed by more than two-fold (fold change, FC) between the groups (−3.84 < logFC > 2.90, *p* < 0.05) (Figure 1, Appendix A). The level of 23 miRNAs (miR-499a-5p, -1255b, -208b, -454, -339-3p, -1468, -485-5p, -320c, -199b-5p, -92a-1*, -411, -99a-5p, -3163, -133a, -99a, -203, -2110, -34c-5p, -450a, -616*, -26b, -144 and let-7g) was lower, and of 12 miRNAs (miR- 339-5p, -335*, -93*, -1270, -873, -214*, -758, -210, -409-3p, -342-5p, -10b, and let-7a*)—higher in HCM. At the same time, only a decrease in the levels of two miRNAs turned out to be significant after adjustment for multiple comparisons: miR-208b (logFC = −3.17, adjusted *p*-value (*p*.adj) = 1.4 × 10^−5^) and miR-499a-5p (logFC = −3.84, *p*.adj = 0.043).

We further analyzed levels of circulating miR-208b and miR-499a-5p in enlarged independent groups of 29 HCM patients and 32 controls, matched by sex and age, using qRT-PCR assay (Figure 2). We failed to validate RNA-seq data on decreased levels of miR-208b and miR-499a-5p in HCM. The level of miR-208b was undetectable in almost all samples (both HCM and controls), whereas the differences in miR-499a-5p level, opposite to the sequencing data, was estimated with an FC equal to 3.4 (*p* = 0.12) in HCM patients compared to controls. Notably, the dispersion in miR-499a-5p levels in the HCM group was much higher than in controls (1.14 × 10^−6^ and 2.85 × 10^−8^, respectively).

To check whether the observed fluctuations of miR-499a-5p levels in the HCM group are caused by specific genetic variants, we estimated levels of miR-499a-5p in patients according to their genotyping status (Figure 3) and observed their genotype-dependent clusterization. 

For further consideration, patients with P/LP variants in *MYH7* gene (*n* = 7) and in *MYBPC3* gene (*n* = 8) were assembled in two separate groups. Patients with P/LP variants in other genes (*TPM1*, *TNNT2*, *TNNI3*, *ALPK3* or *TRIM63*) and patients without identified mutations were merged in the third group. The clinical and demographic characteristics did not differ significantly in these three groups (Appendix A). 

The comparison of miRNA levels between groups of patients showed significant differences; carriers of P/LP variants in the *MYH7* gene were defined with significantly higher level of miR-499a-5p compared to carriers of mutations in the *MYBPC3* gene (FC = 14.1; *p* = 0.0003) and to a combined group of carriers of variants in other genes and patients without identified mutations) (FC = 4.1; *p* = 0.0008) (Figure 4). It could be seen that in *MYH7* variant carriers, the miR-499a-5p levels were also significantly higher than in controls (FC = 8.9; *p* < 0.0001). Conversely, the miR-499a-5p level in *MYBPC3* variant carriers and in the mixed group of other patients did not differ from the controls (Figure 4). 

To study the diagnostic accuracy of miR-499a-5p levels in *MYH7*-positive and *MYH7*-negative patients, the ROC curve was drawn (Figure 5). AUC was 0.95 (95% CI 0.88–1.03, *p* = 0.0004), and sensitivity and specificity were 0.86 and 0.91 (cut-off = 0.0014), respectively.

## 3. Discussion

In our study, high throughput sequencing of circulating miRNAs in HCM patients and individuals without cardiovascular diseases pinpointed miR-499a-5p as one of the dysregulated miRNAs in HCM. Further investigation involving representative groups showed that miR-499a-5p levels are significantly higher in carriers of P/LP variants in the *MYH7* gene than in controls. The mismatch in the direction of changes in miRNA levels according to discovery RNA-seq profiling and validation RT-qPCR data may be due to the genetic heterogeneity of the discovery HCM group. It included patients with mutations in *MYH7* (22%), *TPM1* (22%), *ALPK3* (11%), *MYBPC3* (11%) and *TNNI3* (11%) genes; in two patients (22%), mutations in 19 sequenced genes were not found. Therefore, the small number of *MYH7* variant carriers in the discovery group in combination with the low sensitivity of the high throughput sequencing cast doubt on the significance of results obtained by RNA-seq. As of today, only one study has performed RNA-seq of myocardial tissue from HCM patients with P/LP variants in the *MYH7* and *MYBPC3* genes [22] where the downregulation of miR-487a, miR-654, miR-30d, miR-154, and miR-3193 and upregulation of miR-3671 were reported; differential expression of miR-499 in HCM was not observed.

Further stratification of HCM patients according to genotyping status highlighted that miR-499a-5p was specifically increased in carriers of P/LP variants in the *MYH7* gene compared to its level in carriers of P/LP variants in the *MYBPC3* gene or in other mutations’ carriers and genotype-negative patients that were not identified. Thus, this circulating miRNA could serve as a biomarker of HCM, caused by P/LP variants in the *MYH7* gene. Compared to other miRNAs, namely miR-145-5p, -133a-3p, -143-3p, -26a-5p, -29a-3p, 199a-5p, -27a-3p, -199a-3p, and 155-5p that have been proposed as HCM biomarkers (AUC from 0.71 to 0.89) (reviewed in [21]), miR-499a-5p, according to our findings, has an advantage as a biomarker since it is characterized by AUC = 0.95 and has higher sensitivity and specificity. Moreover, among the aforementioned miRNAs, only miR-499a-5p along with miR-133a exhibited a cardiac-enriched pattern of expression in healthy people that increased the sensitivity of using them as HCM biomarkers [23]. However, in this context, miR-133a demonstrates less predictive accuracy (AUC = 0.71) [16]. Importantly, previous studies did not focus on the genetics of recruited patients. Overall, we do believe that miR-499a-5p as a biomarker may find application in HCM diagnosis, since mutations in the *MYH7* gene are of frequent occurrence in HCM patients (up to 30% of cases), and the miRNA level in plasma is often stable and its analysis is fairly simple.

Our data are directly relevant to the widely studied issue of the association of myosin genes family with HCM and their intergenic relations [24,25,26]. The *MIR499A* gene, encoding miR-499a-5p, is located in intron 19 of the *MYH7B* host gene that produces the β chain of myosin 7b (MYH7B). Besides these products, MYH7B locus was shown to encode a regulatory lncRNA lncMYH7b as well [26]. *MYH7B* established to be an ancient myosin gene that gave the origin to other genes of myosins—*MYH6*, encoding alpha-myosin heavy chain (α-MyHC) and *MYH7*, encoding beta-myosin heavy chain (β-MyHC) [27]. 

Unlike α-MyHC and β-MyHC that are the major sarcomeric myosin proteins expressed in mammalian hearts and determine the contractile velocity of the muscle, the function of MYH7B by now is poorly understood. Interestingly, due to the separate genomic localization of MYH7B locus on chromosome 20 from the other two sarcomeric myosin clusters on chromosomes 14 (cardiac myosins) and chromosome 17 (skeletal muscle myosins), *MYH7B* is believed to have a specialized role in muscle biology in humans [26]. By today, RNA sequencing, western blot and proteomic analyses indicate that full length MYH7b product is not detected (or detected at negligible levels) in human hearts [28,29]. That is explained by the fact that even though MYH7b RNA is transcribed in both heart and skeletal muscles, the majority of its transcripts undergo an alternative splicing event that induces the skipping of exon 8. This leads to a frame-shift event in the transcript’s open reading frame, resulting in the appearance of premature termination codon, which prevents full length protein translation and activates the nonsense-mediated mRNA decay pathway [30]. Several studies indicate that MYH7B has been involved in cardiac function and disease in mammals [28,31,32,33]. The ClinVar database describes a number of LP variants in the *MYH7B* gene that are associated with familial HCM [34]; these variants are not localized in the *MIR499A* gene locus [33].

It is noteworthy that post-transcriptional processes in the *MYH7B* gene do not affect the maturation and expression of the intronic *MIR499A* gene, and miR-499 is highly expressed in the heart [30]. The literature regarding the association of circulating miR-499 with HCM is still limited and contradictory. The level of miR-499a-5p was higher in the plasma of HCM patients than of healthy controls [35]. In another study no significant differences were observed for miR-499-5p when analyzing 41 HCM patients and 41 age- and sex-matched healthy controls [16]. In addition, no differences were found when comparing levels of serum miR-499-5p in HCM patients with and without signs of fibrosis [36]. Notably, in all these studies the genetic analysis of HCM patients was not performed. In another study during profiling of circulating cardiovascular miRNAs, miR-499 was not measurable at all in HCM patients [20]. Additionally, in a study [20] the level of miR-499-5p was undetectable in patients with obstructive and non-obstructive HCM and aortic stenosis; the authors didn’t provide information on the levels of this miRNA in the control group. The examination of miR-499 in the serum of 40 healthy controls and eight patients carrying *MYH7B* potential pathogenic variants did not find significant changes in its level [33]. 

Only a few studies investigated the functional role of miR-499. Thus, mice with miR-499 and miR-208b (intronic miRNA transcribed from another myosin host gene, *Myh7*) were shown to regulate cardiac β-MyHC/α-MyHC ratio. At the same time, the expression of these miRNAs is regulated by miR-208a (intronic miRNA transcribed from *Myh6* host gene) [37]. However, it should be noted that the adult mouse cardiac β-MyHC/α-MyHC ratio is opposite to the human one (5%:95% compared with 90%:10%), demonstrating species-specific regulation [24,38], therefore the investigation of the regulatory role of miR-499a-5p on myosin action requires experiments in humans. In the cellular model of human induced pluripotent stem cells differentiated to cardiomyocytes, which were treated with an anti-miR that specifically targeting miR-499, no changes in *MYH7B* or β-MyHC levels were observed, confirming that miR-499 does not regulate β-MyHC expression in human cardiomyocytes [26]. Further investigation is needed to uncover the regulatory role of miR-499 in the heart. 

Intriguingly, MYH7b RNA levels were shown to correlate with β-MyHC expression, including the known increase in β-MyHC expression in human hearts from patients with ischemic cardiomyopathy [26]. The β-MyHC induction, which shifts the β-MyHC/α-MyHC ratio to ~100% β-MyHC, was also observed in other heart diseases [39,40,41]. Moreover, point mutation R723G introduced in the *MYH7* gene in the neonatal pigs resulted in the appearance of HCM features, including mild myocyte disarray, malformed nuclei, and *MYH7* overexpression [42]. A significant cell-to-cell variation was reported in mutated vs. wild-type MYH7 mRNA expression in individual cardiomyocytes isolated from the same tissue samples of HCM patients with two different heterozygous β-MyHC-mutations; this strong variability is most likely due to stochastic, burst-like *MYH7* transcription, which is independent for the mutant and the wild-type allele [43]. Based on these findings we hypothesize that mutations in the *MYH7* gene may drive *MYH7* transcription in the heart followed by the increased expression of the *MYH7B* gene and, as a result, *MIR499A* overexpression, which could be further actively or passively secreted from the cardiomyocytes into the bloodstream. 

## 4. Materials and Methods

### 4.1. Subjects

The discovery group included nine HCM patients (females 55.5%, mean age 50.2 ± 10.3 years); the independent validation group consisted of 29 HCM patients (females 41%, mean age 48.0 ± 12.6 years). The clinical and demographic characteristics of patients from these groups are presented in Table 1. There were no differences in described characteristics between two groups. All patients were previously studied by a targeted massively parallel sequencing method using a panel that includes genes related to hypertrophic cardiomyopathy or its main phenocopies (*ACTC1*, *ALPK3*, *DES*, *FHL1*, *FHOD3*, *GLA*, *LAMP2*, *MYBPC3*, *MYH7*, *MYL2*, *MYL3*, *PRKAG2*, *PTPN11*, *TNNC1*, *TNNI3*, *TNNT2*, *TPM1*, *TRIM63*, *TTR*) at accredited genetics laboratories. Patients were diagnosed with HCM based on 2014 criteria of the European Society of Cardiology [44]: the presence of increased LV wall thickness ≥ 15 mm that was not solely explained by abnormal loading conditions. Patients suspicious of phenocopies of HCM were not included in the study. Exclusion criteria were also systemic autoimmune, viral, bacterial and oncological diseases. Control groups included individuals without cardiovascular diseases and consisted of 6 persons (females 50%, mean age 31.3 ± 5.9 years) for RNA-seq analysis and 32 persons (females 50%, mean age 46.5 ± 12.1 years) for validation analysis. 

### 4.2. Plasma Collection and RNA Extraction

Blood samples were collected from the subjects in EDTA-containing tubes. Within 2 h the tubes with whole blood were centrifuged at 1000× *g* for 10 min at room temperature. Platelet-free plasma was obtained by additional double centrifugation at 2500× *g* for 15 min. The isolated plasma in 0.2 mL aliquots was frozen at −80 °C prior to use. Plasma samples were thawed at room temperature and centrifuged at 2000× *g* to remove debris. Plasma miRNAs were isolated using a miRNeasy Serum/Plasma Kit (Qiagen, Düsseldorf, Germany) according to the manufacturer’s protocol.

### 4.3. Small RNA Sequencing Libraries Preparation

The construction of small RNA libraries was performed using TruSeq^®^ Small RNA Library Prep Kit (Illumina, San Diego, CA, USA) according to the manufacturer’s instructions with minor modifications. 3′ and 5′ RNA adapters were ligated to miRNA which was further reverse-transcribed into cDNA. The amplified cDNA libraries (18 PCR cycles) were purified using a Qiaquick PCR Purification kit (Qiagen, Düsseldorf, Germany) and then loaded on a 6% Novex TBE gel (ThermoFisher Scientific, Waltham, MA, USA). 145–160 bp cDNA fragments were excised and eluted from the gel, precipitated and dissolved in 12 μL of TE-buffer. The purity and size of the cDNA libraries were checked on a QIAxcel Advanced System using a High Sensitivity DNA chip (Qiagen, Düsseldorf, Germany) and quantified using a Quantus Fluorometer (Promega, Madison, WI, USA). Each sample was individually barcoded and sequenced on an Illumina MiSeq platform in multiplexed pools (three samples per sequencing run) following the 36 bp fragment protocol.

### 4.4. Small RNA-seq Data Analysis

All sequencing reads were demultiplexed using CASAVA (Illumina, San Diego, CA, USA). The low quality nucleotides and adapter sequences were trimmed from fastq files using Trimmomatic [45]. The trimmed reads were aligned and annotated to miRBase version 21 (http://www.mirbase.org/ accessed on 18 December 2021) using Bowtie2 [46]. The differences in expression levels were calculated using the package edgeR version 3.14.0 [47]. The libraries were normalized using the Relative Log Expression method [48]. To detect differential expression modules a generalized linear model method was used [49]. Technical variations as known blocking factors were included in the quasi-likelihood negative binomial generalized log-linear model [50]. The False Discovery Rate (FDR) procedure of Benjamini-Hochberg was used to adjust for multiple testing corrections (*p*.adj < 0.05 were considered significant) [51]. 

### 4.5. RT-qPCR

The input miRNAs were reverse transcribed using the TaqMan microRNA Reverse Transcription Kit (ThermoFisher Scientific, Waltham, MA, USA) and cDNA was then subjected to qPCR in triplicate using the TaqMan miRNA Assays (ThermoFisher Scientific, Waltham, MA, USA) according to the manufacturer’s protocol. Cel-miR-39-3p was used as an endogenous control due to its stable levels in all the samples. The MiRNAs differential expression was calculated using the Delta-Delta Ct method, where ΔΔCt for each miRNA = ((average of Ct minus average of Ct cel-miR-39-3p in the MI group) minus (average of Ct minus average of Ct cel-miR-39-3p in the healthy control group)) [52]. 

### 4.6. Statistical Analysis

Continuous variables were tested for normality of distribution using D’Agostino & Pearson omnibus and Shapiro-Wilk normality tests. Variables with approximately normal distribution are presented as mean ± standard deviation (SD), and those with skewed distribution—as median and interquartile range. Continuous variables were compared between groups using independent *t*-test. Categorical variables were evaluated using Fishers’ exact test. The comparison of miRNA levels between different groups was performed by a Mann-Whitney U test (*p* < 0.05 were considered significant). All analyses and graphical representation were performed in GraphPad Prism v. 7.0 (GraphPad Software, San Diego, CA, USA). 

### 4.7. ROC-Curve Analysis

The diagnostic value of miRNA was evaluated from the area under the curve (AUC) in the receiver operating characteristic (ROC) analysis using GraphPad Prism v.7.0 (GraphPad Software, San Diego, CA, USA). The optimal cut-off value, sensitivity, and specificity were determined by calculating the Youden index. *p* < 0.01 was assumed to be statistically significant.

## 5. Conclusions

Overall, relying on our findings we suggest that miR-499a-5p could serve as a circulating biomarker of HCM, caused by mutations in the *MYH7* gene. Our study highlights that miRNA expression, and, therefore, post-transcriptional gene regulation, is sensitive to the genotyping status of HCM patients. Further investigations need large homogeneous samples of genetically characterized HCM patients to elucidate the role of miR-499a-5p in the development of HCM. They will rely on the evaluation of their appropriateness as biomarkers for the disease. Another issue is to determine whether miR-499a-5p levels in plasma reflects the cardiac damage or whether it is secreted actively or passively from the cells. It is also important to understand whether this miRNA might play a role in intercellular signaling.

## Figures and Tables

**Figure 1 ijms-23-03791-f001:**
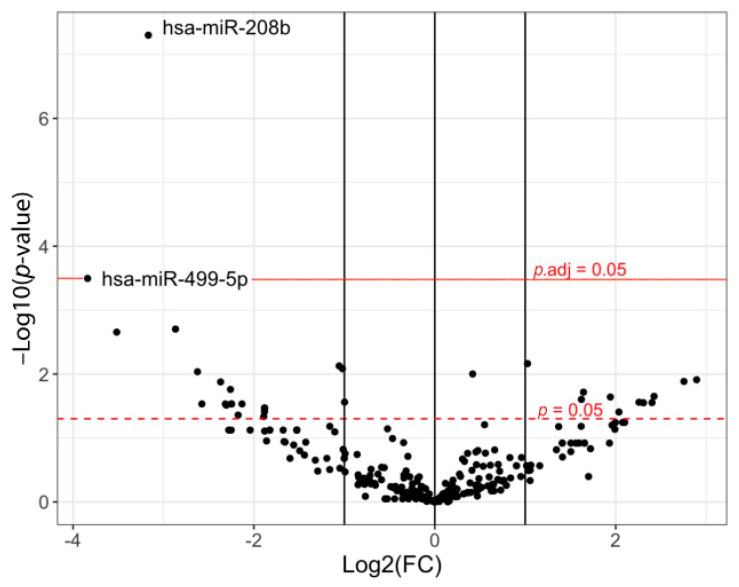
The comparison of the profiles of circulating miRNAs in patients with HCM (*n* = 9) and controls (*n* = 6). On the graph, the *X*-axis represents the values of the fold change (FC) in miRNA levels, the *Y*-axis represents the *p* values, all in a logarithmic scale. The dotted line indicates threshold for statistical significance *p* = 0.05; solid line—threshold for statistical significance adjusted *p*-value (*p*.adj) = 0.05; dots—miRNAs.

**Figure 2 ijms-23-03791-f002:**
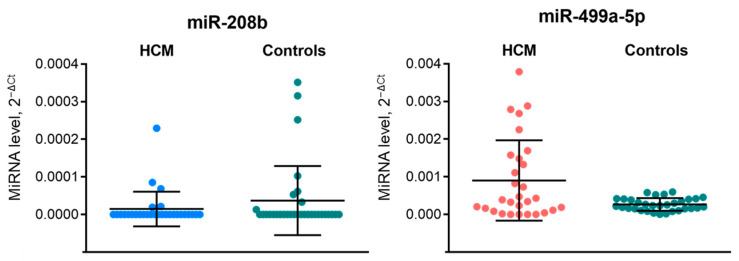
Plasma levels of miR-208b and miR-499a-5p in 29 HCM patients and 32 healthy controls as determined by RT-qPCR. Here and below, miRNA levels were calculated relative to cel-miR-39-5p using the delta-delta Ct method. Data is presented using a scatter plot (mean with SD). Differences in miRNA levels between HCM patients and healthy controls were insignificant (*p* > 0.05) according to a Mann-Whitney U test. The green dots define levels of miR-208b and miR-499a-5p in controls, blue dots and red dots—levels of miR-208b and miR-499a-5p, respectively, in patients with HCM. HCM—hypertrophic cardiomyopathy.

**Figure 3 ijms-23-03791-f003:**
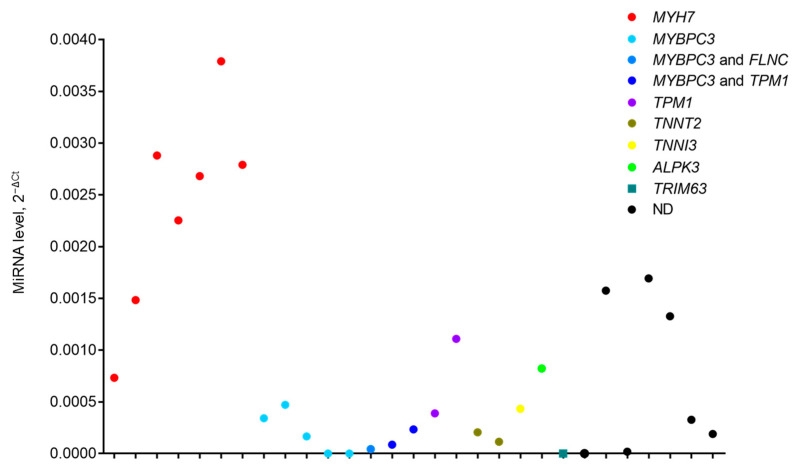
Level of circulating miR-499a-5p in each HCM patient depending on sequencing data of HCM-associated genes. ND—patients in whom mutations in sequenced genes were not determined.

**Figure 4 ijms-23-03791-f004:**
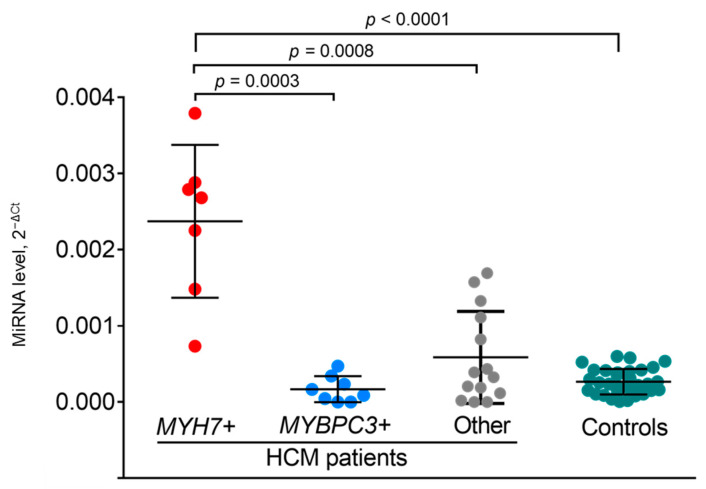
Levels of miR-499a-5p in carriers of mutations in *MYH7* gene (*n* = 7), in carriers of mutations in *MYBPC3* gene (*n* = 8), in mixed group of non-*MYH7* and non-*MYBPC3* mutations carriers and of genotype-negative patients (*n* = 14); and controls (*n* = 32) as determined by RT-qPCR. Significant differences were determined by using a Mann-Whitney U test.

**Figure 5 ijms-23-03791-f005:**
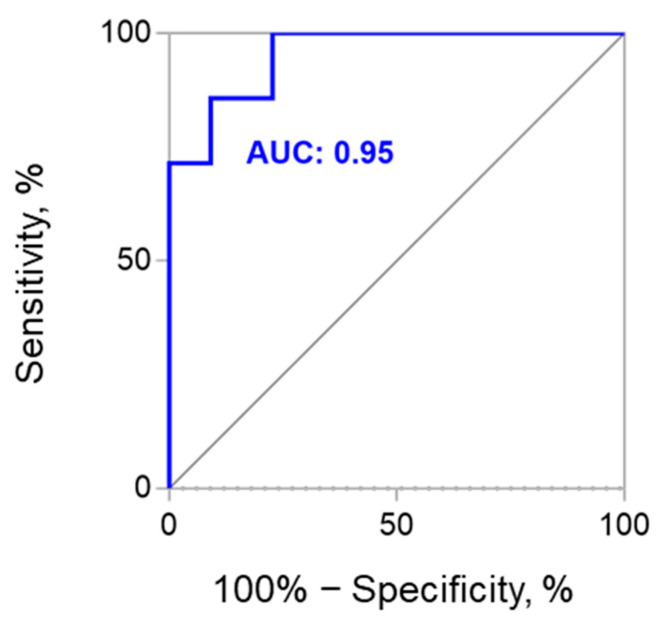
The ROC curve analysis of circulating miR-499a-5p for detecting *MYH7*-positive HCM patients. ROC—receiver operating characteristic; AUC—area under the curve.

**Table 1 ijms-23-03791-t001:** Clinical characteristics of HCM patients involved in the study (*n* = 29).

Characteristics	Discovery Group	Validation Group
Female, *n* (%)	5 (55.5)	12 (41)
Age, years	50.2 ± 10.3	48.0 ± 12.6
BMI, kg/m2	26.69 ± 4.73	26.95 ± 4.14
Five-year HCM Risk-SCD (%)	5.24 ± 2.94	3.87 ± 2.30
Family history of HCM, *n* (%)	4 (44.4)	12 (41.4)
Family history of SCD, *n* (%)	2 (22.2)	5 (17.2)
Atrial fibrillation, *n* (%)	1 (11.1)	4 (13.8)
Ventricular tachycardia, *n* (%)	4 (44.4)	6 (20.7)
Arterial hypertension, *n* (%)	3 (33.3)	12 (41.4)
Coronary heart disease, *n* (%)	1 (11.1)	2 (6.9)
Diabetes mellitus, *n* (%)	0 (0)	1 (3.4)
Echocardiography		
Maximal LVWT, mm	25.50 ± 6.41	23.56 ± 5.54
Left atrial diameter, mm	42.11 ± 5.30	44.00 (41.00–47.00)
LA ESV index, mL/m^2^	45.84 ± 13.45	44.39 ± 10.47
LV EDD index, mm/m^2^	22.30 (21.80–24.85)	23.57 ± 2.20
LV ESD index, mm/m^2^	15.10 (9.70–16.50)	13.90 (10.04–16.00)
LV EDV index, mL/m^2^	47.88 ± 13.97	44.41 ± 11.78
LV ESV index, mL/m^2^	16.99 ± 8.66	13.10 (10.80–17.60)
LVOT obstruction, *n* (%)	5 (55.5)	9 (31.0)
LVEF (%)	65.89 ± 8.25	67.86 ± 6.77
Mitral E-e’ ratio	14.15 ± 9.50	9.30 (6.35–14.3)
Apical form of HCM, *n* (%)	0 (0)	5 (17.2)
Electrocardiography		
Pathological Q Waves, *n* (%)	4 (44.4)	7 (24.1)
T-Wave inversion, *n* (%)	8 (88.9)	22 (75.9)
Sokolow-Lyon index, mm	32.11 ± 16.87	31.00 (25.00–41.00)

BMI—body mass index, HCM—hypertrophic cardiomyopathy, SCD—sudden cardiac death, LVWT—left ventricular wall thickness, LA—left atrial; ESV—end-systolic volume, EDD—end diastolic diameter, ESD—end-systolic diameter, EDV—end-diastolic volume, ESV—end-systolic volume, LVOT—left ventricular outflow tract; LVEF—left ventricular ejection fraction; E/e’—early transmitral flow velocity to early mitral annular tissue velocity to estimate LV filling pressure.

## Data Availability

MiRNA sequencing data are deposited in the international public repository, Gene Expression Omnibus, under accession identification as GSE195436.

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
