# Peer review of "Circulating miR-499a-5p Is a Potential Biomarker of MYH7—Associated Hypertrophic Cardiomyopathy"

_ijms, 2022, doi:10.3390/ijms23073791_

Round 1
Reviewer 1 Report
The paper by Baulina and colleagues depicts the usefulness of circulating miRNA profiling as a biomarker tool for MYH7-associated hypertrophic cardiomyopathy. Although the study subject is an interesting one, it needs further clarification and research.
Major comments:
1) The contradictory results obtained in NGS and RT-PCR need further clarification. Considering the explanation the authors offer, one could hypothesize that if the patient groups were to be extended (more patients, different mutations) then the results obtained could be very distinct.
2) Following the previous comment if the authors want to correlate circulating miRNAs levels with the stratification of HCM patients according to genotyping status, it would be wise to enlarge the studied population. The number of patients enrolled is scarce when evaluating biomarkers and extremely insufficient if we are considering genetic variance.
3) The role of miRNA based biomarkers in cardiovascular diseases, in particular in HCM could be more developed in the introduction and discussion.
Reviewer 2 Report
In this work the authors, starting from a high-throughput miRNAs sequencing approach followed by a validating experiment in HCM patients and control samples propose miR-499a-5p as biomarker for MYH7-associated HCM. The work is interesting, but it needs some experimental clarification to better explain the experimental flow of the scientific hypothesis. The experimental data presented in this work are consistent, the procedures are of high standards, and the paper is well written.
Below I list my suggestions and my clarification requests.
Major Revisions:
The major point, that should be clarified in the work, is the opposite results obtained in the two miRNAs expression analysis (RNA sequencing and RT-PCR) on the differential expression of miR-499a-5p between HCM and control samples. In fact, in the first analysis it resulted downregulated in HCM patients, instead in the RT-PCR experiment it was up-regulated in MYH7-HCM patients.
The first point, not so clear in the text, is whether the two cohorts of patients (used in the two approaches) are different and independent samples. Probably the discrepancy is due to a too small amount of patients using to perform the miRNAs sequencing. This pitfall could also explain the lack of other miRNAs differentially expressed between HCM patients and control samples with a significant FDR p value. It could be useful to enlarge the analysis of sequencing also for the other samples. In addition, I think that is necessary to analyse the genetic variants profile of genes related to HCM of the 9 HCM patients used in the first analysis. Are patients with MYH7 genetic variants included? If not, the two analysis are not comparable.
Minor Revisions:
- I suggest to include a table of general clinical characteristics also for the cohort of patients used in miRNAs sequencing analysis.
- There are some sentences that should be better supported by references:
- Lines 38 and 40: references for genetic variants related to HCM should be added.
- Line 141-142: this sentence should be supported by references.
- I think that in the discussion section, it should be important to report previous published data obtained through miRNAs sequencing, citing some papers using RNA-seq and discussing their results.
- In material and methods, I suggest to add a dedicated paragraph for computational and statistical analysis and for the prediction model. These two experimental procedures should be more detailed.
Round 2
Reviewer 1 Report
The second version of the paper by Baulina and colleagues is a much-improved report on the impact of circulating miRNAs as biomarkers for the MYH7-HCM condition. The alterations made concerning the presentation and discussion of the results contribute greatly to a better comprehension of the author's findings, despite the mismatch in the miRNAs levels. I acknowledge the authors' difficulty to scale up their data which is now properly addressed in the conclusions. Overall, the presentation of the research is now significantly improved.
Reviewer 2 Report
Although the opposite results obtained in the discovery and validation miRNAs expression analysis, I think that now the work completely discuss its reported results and underline its limits. Surely, to perform extra miRNA sequencing for other HCM patients can improve the results of the paper, but I understand the financial and logistic problems highlighted by the authors. Moreover, I think that now the methods and scientific approaches are sufficiently detailed.